# High-Intensity Interval Training as Redox Medicine: Targeting Oxidative Stress and Antioxidant Adaptations in Cardiometabolic Disease Cohorts

**DOI:** 10.3390/antiox14080937

**Published:** 2025-07-30

**Authors:** Dejan Reljic

**Affiliations:** 1Department of Medicine 1—Gastroenterology, Pneumology and Endocrinology, University Hospital Erlangen, Friedrich-Alexander University Erlangen-Nürnberg, Ulmenweg 18, 91054 Erlangen, Germany; dejan.reljic@uk-erlangen.de; Tel.: +49-9131-85-40116; 2Hector-Center for Nutrition, Exercise and Sports, Department of Medicine 1, University Hospital Erlangen, Friedrich-Alexander University Erlangen-Nürnberg, Ulmenweg 18, 91054 Erlangen, Germany; 3Deutsches Zentrum Immuntherapie (DZI), University Hospital Erlangen, Friedrich-Alexander University Erlangen-Nürnberg, Ulmenweg 18, 91054 Erlangen, Germany

**Keywords:** HIIT, oxidative stress, antioxidant capacity, cardiometabolic disease, redox balance, exercise therapy

## Abstract

High-intensity interval training (HIIT) has emerged as a promising non-pharmacological intervention for improving cardiometabolic health. In populations with diabetes, cardiovascular disease, obesity, or metabolic dysfunction, redox imbalance—characterized by elevated oxidative stress and impaired antioxidant defense—is a key contributor to disease progression. This narrative review synthesizes current evidence on the effects of HIIT on oxidative stress and antioxidant capacity across diverse cardiometabolic disease cohorts. While findings are heterogeneous, the majority of studies demonstrate that HIIT intervention can reduce levels of oxidative stress markers and enhance antioxidant enzyme expression. These redox adaptations may underpin improvements in vascular endothelial function, inflammation, and metabolic regulation. Importantly, variations in intensity, duration, and health status influence these responses, highlighting the need for individualized exercise prescriptions. Safety considerations are emphasized, including the necessity for medical clearance, gradual progression, and individualized training prescriptions in higher-risk individuals. In conclusion, HIIT shows potential as a targeted strategy to restore redox homeostasis and improve cardiometabolic outcomes, although further research is needed to clarify optimal protocols and the underlying mechanisms.

## 1. Introduction

Oxidative stress, defined as an imbalance between the production of reactive oxygen species (ROS) and the body’s antioxidant defense systems, plays a pivotal role in the pathogenesis and progression of chronic non-communicable diseases, including cardiometabolic disorders such as type 2 diabetes mellitus (T2DM) [1], cardiovascular disease (CVD) [2], and metabolic syndrome [3]. Under these conditions, persistent oxidative stress contributes to endothelial dysfunction, mitochondrial impairment, systemic inflammation, and insulin resistance—core mechanisms driving disease onset and exacerbation. Elevated levels of oxidative biomarkers such as malondialdehyde (MDA), protein carbonyls, and 8-hydroxy-2′-deoxyguanosine (8-OHdG) are consistently reported in various clinical populations [4].

Among non-pharmacological interventions, lifestyle modifications, particularly regular physical exercise, are foundational in preventing and managing cardiometabolic diseases. Exercise functions as a physiological stressor that can induce beneficial redox adaptations when applied appropriately by upregulating endogenous antioxidant defense systems and improving cellular resilience [5]. Among contemporary exercise modalities, high-intensity interval training (HIIT), which alternates short bursts of vigorous effort with periods of rest or low-intensity activity, has gained particular attention in recent years for its time efficiency and therapeutic efficacy [6]. Evidence suggests that HIIT can improve a wide range of health outcomes, including cardiovascular function [7], low-grade inflammation [8], insulin sensitivity [9], and quality of life [10] in clinical cohorts. 

However, the impact of HIIT on oxidative stress regulation remains insufficiently understood. While some studies report beneficial antioxidant adaptations, others raise concerns regarding pro-oxidant responses, especially in individuals with compromised redox homeostasis [11,12,13]. Moreover, questions remain about the optimal dose, safety, and patient suitability of HIIT in disease contexts where oxidative stress is already elevated. In view of these uncertainties and knowledge gaps, it is timely to shed light on the conceptual framing of HIIT as a tool within redox medicine and to critically discuss its potential to alter redox signaling pathways for therapeutic benefit. Therefore, this narrative review aims to (i) synthesize recent findings on HIIT-induced redox responses in cardiometabolic disease cohorts, (ii) identify gaps and controversies in the literature, (iii) address practical recommendations, and (iv) provide a forward-looking evaluation of the potential of HIIT as a non-pharmacological redox-modifying intervention within cardiometabolic disease settings. 

## 2. Oxidative Stress in Cardiometabolic Diseases

### 2.1. The Pathophysiological Role of Oxidative Stress

Oxidative stress, arising from an imbalance between ROS production and the body’s antioxidant defenses, plays a central role in the pathophysiology of cardiometabolic diseases [1,2,3]. Major ROS, such as superoxide anions, hydrogen peroxide (H_2_O_2_), and hydroxyl radicals, are generated during normal cellular metabolism, particularly within the mitochondria. Under physiological conditions, these molecules function in redox signaling, modulating vascular tone, immune responses, and cellular proliferation. However, when produced in excess or inadequately neutralized, ROS can damage cellular lipids, proteins, and nucleic acids, thereby contributing to tissue dysfunction and disease progression [4]. 

In the vascular system, for example, oxidative stress reduces the bioavailability of nitric oxide (NO) through direct interaction with superoxide, impairing endothelial function and promoting vasoconstriction, thrombosis, and inflammation. Moreover, ROS-mediated lipid peroxidation products such as MDA and 4-hydroxynonenal modify cellular structures and initiate atherogenic processes. In parallel, oxidative modifications of insulin receptor substrates disrupt insulin signaling, thereby promoting insulin resistance. Additionally, chronic ROS elevation has been reported to activate pro-inflammatory pathways, including nuclear factor-kappa B (NF-κB) and mitogen-activated protein kinases, which further exacerbate metabolic dysfunction. Mitochondrial DNA damage caused by oxidative stress also leads to decreased ATP production and metabolic inflexibility, especially in insulin-sensitive tissues [14]. 

### 2.2. Oxidative Stress as a Mechanistic Player in Disease Development

In cardiometabolic diseases, oxidative stress is not merely a consequence of pathology but also acts as a driver of disease progression. In hypertension, for example, excessive ROS levels impair endothelium-dependent vasodilation by uncoupling endothelial nitric oxide synthase (eNOS) and increasing vascular tone. This contributes to structural and functional changes in the arterial wall, leading to increased peripheral resistance [15]. In T2DM, oxidative stress disrupts glucose homeostasis by interfering with insulin signaling. Specifically, ROS stimulate serine phosphorylation of insulin receptor substrate-1, inhibiting downstream insulin action. In pancreatic β-cells, which possess relatively low antioxidant capacity, excessive ROS can impair insulin secretion and induce apoptosis, contributing to β-cell dysfunction [1]. In the context of atherosclerosis, ROS promote the oxidation of low-density lipoproteins (LDL), facilitating foam cell formation and plaque development. Additionally, the oxidative microenvironment enhances the expression of adhesion molecules and cytokines that promote leukocyte infiltration and chronic inflammation within vascular walls [16]. 

Excessive ROS generation can also directly impair mitochondrial function by damaging mitochondrial DNA, lipids, and respiratory chain proteins, leading to decreased ATP production, increased electron leakage, and further ROS generation, a vicious cycle that contributes to metabolic inflexibility and cellular energy deficits [16]. Such mitochondrial dysfunction is particularly detrimental in insulin-sensitive tissues like skeletal muscle and the heart, where it exacerbates insulin resistance and impairs contractile function. Taken together, oxidative stress contributes to nearly all stages of cardiometabolic disease development—from endothelial dysfunction and insulin resistance to mitochondrial impairment and inflammation. 

In this context, a critical and ongoing question is whether oxidative stress represents a primary driver or a secondary consequence of disease pathology. The available evidence suggests a bidirectional relationship, whereby oxidative stress both arises from metabolic dysregulation and, in turn, exacerbates it through self-reinforcing mechanisms [3,17]. This dual role complicates the interpretation of redox biomarkers and presents a challenge for therapeutic targeting. As such, strategies aimed at restoring redox homeostasis must consider not only the suppression of excess reactive species but also the preservation of physiological ROS signaling required for cellular adaptation and metabolic regulation. Readers seeking a more detailed discussion and visual representations of the molecular mechanisms underlying oxidative stress at the cellular level in cardiometabolic diseases are referred to previous comprehensive reviews [1,2].

### 2.3. Elevated Oxidative Stress in Clinical Populations

Elevated levels of oxidative stress markers are consistently observed in individuals with cardiometabolic diseases. For instance, patients with metabolic syndrome typically show increased circulating levels of lipid peroxidation markers such as MDA and thiobarbituric acid reactive substances (TBARS), along with decreased activities of enzymatic antioxidants like superoxide dismutase (SOD), catalase (CAT), and glutathione peroxidase (GPx) [3,18]. In individuals with obesity and insulin resistance, levels of both systemic and tissue-specific markers of oxidative damage are elevated. Studies have shown that obese individuals exhibit higher basal mitochondrial H_2_O_2_ emission rates in skeletal muscle, coupled with reduced glutathione (GSH) levels and SOD activity [19]. In T2DM, oxidative DNA damage, as indicated by increased urinary 8-OHdG levels, is a reliable marker of systemic oxidative burden [20]. Similarly, in hypertensive and CVD populations, decreased antioxidant enzyme activities have been associated with arterial stiffness and endothelial dysfunction. In these patients, oxidative imbalance contributes to the progression of vascular damage, ultimately increasing the risk of myocardial infarction and stroke [2,15]. 

While exogenous antioxidant therapies, such as targeted supplementation with vitamins C and E, polyphenols, or synthetic scavengers, have shown some promise in preclinical models, their clinical efficacy in reducing cardiovascular or metabolic disease risk remains inconclusive. Systematic reviews have reported only minimal or no significant impact on cardiometabolic endpoints [21,22,23], and concerns have been raised regarding the potential blunting of beneficial ROS-mediated signaling cascades essential for metabolic and muscular adaptations [24]. These limitations have shifted attention toward interventions that enhance the body’s endogenous antioxidant capacity rather than simply neutralizing ROS. In this context, physical exercise has emerged as a promising strategy for inducing redox adaptations through hormetic mechanisms.

## 3. The Role of Exercise in the Prevention and Treatment of Cardiometabolic Disorders with Focus on HIIT

Physical exercise is universally recognized as a frontline strategy in both the prevention and management of cardiometabolic disorders, including CVD, T2DM, hypertension, and metabolic syndrome [25,26]. Numerous large-scale cohort and interventional studies have demonstrated that regular physical activity and exercise can significantly attenuate cardiometabolic risk by improving glycemic control, blood pressure, lipid profiles, body composition, and vascular function—independent of weight loss [27]. The underlying mechanisms are multifactorial and include improved insulin sensitivity, enhanced mitochondrial function, increased endothelial nitric oxide production, as well as reductions in systemic inflammation and oxidative stress [5,28].

Historically, cardiovascular-type exercise, in particular moderate-intensity continuous aerobic training (MICT), has been advocated as the predominant exercise intervention for individuals with cardiometabolic disorders and obesity. Additionally, strength training (ST) is also widely recognized as an essential component of a comprehensive exercise regimen—offering benefits such as improved muscular strength, glucose control, and insulin sensitivity [29,30]. Thus, guidelines from major health organizations recommend a combination of both cardiovascular and muscle-strengthening training for optimal outcomes in chronic disease prevention and management [31]. Due to the specific focus of this article, the effects of strength-based exercise will not be discussed here. 

In recent years, more time-efficient exercise strategies, in particular various types of HIIT, have gained increasing attention as an alternative or adjunct to traditional training approaches in the general population as well as in clinical settings [6]. HIIT involves repeated bouts of short-duration, vigorous-intensity exercise interspersed with periods of active or passive recovery. Despite reduced total time commitment, HIIT has been shown to yield equal or even superior improvements in key cardiometabolic parameters when compared to MICT [9,32,33]. Importantly, clinical studies have demonstrated the safety and feasibility of HIIT in diverse patient populations, including individuals with obesity, metabolic syndrome, CVD, and T2DM, when appropriately prescribed and monitored [32,33,34,35], challenging earlier assumptions that such training might be too intense for clinical cohorts. 

Apart from the well-established benefits of regular exercise, such as enhanced cardiorespiratory fitness, insulin sensitivity, or vascular health [25,26], there is increasing recognition that exercise-induced modulation of oxidative stress and antioxidant defense systems play key mechanistic roles in improving cardiometabolic health [5]. However, the relationship between exercise and oxidative stress is complex and appears to depend on several factors, particularly exercise volume, intensity, and individual health status. The following section will summarize the current knowledge on the acute and long-term/adaptive effects of HIIT and discuss how intensity, volume, and health status may affect these responses.

## 4. HIIT and Oxidative Stress

### 4.1. Acute Effects of HIIT on Oxidative Stress Status

Vigorous-intensity exercise, particularly when performed at or above ~75% of an individual’s maximal oxygen uptake (VO_2max_), is known to acutely increase the generation of ROS, thereby promoting a transient state of oxidative stress. Importantly, this increase is a normal physiological response to elevated metabolic demands and mitochondrial respiration, as well as enhanced activities of ROS-generating enzymes such as nicotinamide adenine dinucleotide phosphate oxidase (NOX) and uncoupled nitric oxide synthase [12]. During intense exercise, various body tissues—especially skeletal muscle and the vascular endothelium—are exposed to elevated oxygen flux, leading to a rise in electron leakage from the mitochondrial electron transport chain, which further amplifies superoxide production [36]. 

The magnitude of this acute oxidative response is strongly influenced by both exercise intensity and volume. Higher intensity elicits greater oxygen consumption per unit of time, thereby producing more ROS during exercise [37]. Longer duration of a training session can extend this oxidative load, potentially overwhelming endogenous antioxidant systems [37]. For example, prolonged endurance exercise (e.g., long-distance running or high-volume cycling) has been shown to significantly elevate levels of markers of lipid peroxidation (e.g., MDA) and DNA oxidation (e.g., 8-OHdG) for up to 24 h post-exercise, especially in untrained or poorly adapted individuals [12,38]. These findings underscore that both intensity and duration are critical in shaping the oxidative response. Moreover, the individual’s training status and health condition modulate this effect. In sedentary or clinical populations, even moderate-to-vigorous intensity exercise can potentially produce disproportionate increases in levels of oxidative stress markers due to reduced baseline antioxidant capacity [37]. Conversely, in well-trained individuals, the same oxidative challenge may be buffered more effectively due to prior upregulation of enzymatic and non-enzymatic antioxidant defenses.

By design, HIIT involves short bursts of vigorous exercise, typically at 85–95% of maximal heart rate (HR_max_). This exercise intensity can elicit a marked acute increase in oxidative stress, similar to that observed in other forms of intense exercise. Studies have consistently reported elevated levels of oxidative biomarkers such as MDA, protein carbonyl, and 8-OHdG immediately after a single HIIT session in healthy individuals [11,13]. At the same time, short HIIT protocols typically involve a relatively low total exercise volume, which may limit the duration of oxidative exposure and reduce the risk of excessive damage compared to higher-volume endurance sessions. This acute ROS surge is not pathological per se; rather, it serves as a compelling physiological signal. This phenomenon—often referred to as “oxidative eustress”—triggers signaling pathways, including the activation of NF-κB and nuclear factor erythroid 2–related factor 2 (Nrf2) transcription factors. These pathways enhance the synthesis of endogenous antioxidants like SOD, CAT, and GPx [12]. Studies in healthy young adults performing a single HIIT session have demonstrated rises in ROS markers and transient dips in antioxidant enzyme activities within hours post-exercise. These shifts typically return to baseline within 24–48 h in well-trained individuals with robust physiological resilience [11].

However, clinical populations may present unique considerations. It has been reported, for example, that the acute oxidative stress response to vigorous exercise appears more pronounced in obese compared to non-obese individuals [39]. Patients with cardiometabolic conditions typically exhibit elevated baseline oxidative stress and impaired antioxidant systems [18], making them potentially more susceptible to ROS amplification during intense exercise. These findings highlight the need for careful monitoring and individualized training prescription in cardiometabolic disease cohorts. Importantly, while this acute rise in oxidative stress might raise concerns about potential harm, it may also act as a necessary stimulus for longer-term adaptive benefits—if applied appropriately. Understanding how these acute responses translate into chronic adaptations is essential for the safe and effective implementation of HIIT in cardiometabolic care.

### 4.2. Longer-Term Effects of HIIT on Oxidative Stress Status

While acute bouts of HIIT transiently elevate ROS and markers of oxidative damage, consistent exposure to this oxidative challenge appears to elicit beneficial adaptations in the body’s endogenous antioxidant systems. This biphasic or hormetic response, where a temporary stressor induces longer-term resilience, is a cornerstone of exercise physiology and particularly relevant in the context of oxidative stress modulation. 

In healthy individuals, regular HIIT has been shown to upregulate the activities and expression of key antioxidant enzymes, such as SOD, CAT, and GPx, as well as non-enzymatic antioxidants like GSH. These adaptations help to improve redox homeostasis and reduce resting levels of oxidative damage. For example, Bogdanis et al. [40] observed that a short-term HIIT program (4 × 30-s cycle-ergometer sprints, 3 sessions/week) led to significant increases in CAT activity in skeletal muscle in healthy physically active men after only 3 weeks. Similarly, Costa et al. [41] found that a 4-week progressive HIIT program (8–12 × 1 min cycling bouts at 90–110% peak power, 3 sessions/week) significantly increased plasma total antioxidant capacity and erythrocyte catalase activity while reducing lipid peroxidation in healthy young men, indicating improved systemic redox homeostasis. These adaptive responses are believed to occur via redox-sensitive signaling pathways. ROS generated during high-intensity exercise transiently activate transcription factors such as Nrf2, which controls the expression of a suite of cytoprotective genes involved in antioxidant defense, mitochondrial biogenesis, and anti-inflammatory responses [42]. Notably, these benefits can be achieved with relatively low training volumes. In a study by Gibala et al. [43], healthy participants completing just three 20-min sessions of HIIT per week (consisting of 4–6 × 30-s all-out cycling sprints) demonstrated similar improvements in oxidative capacity and antioxidant enzyme expression as those engaging in traditional higher-volume endurance training. This suggests that vigorous, very low-volume exercise may be an effective and time-efficient strategy to induce benefical antioxidant adaptations in healthy populations.

In recent years, HIIT has also been increasingly studied as a potential therapeutic modality to modulate oxidative stress in individuals with cardiometabolic disease. The findings of these trials are disscussed in the following sections and summarized in Table 1.

#### 4.2.1. Type 2 Diabetes Mellitus

To date, most of the studies investigating the effects of HIIT on oxidative stress status in cardiometabolic cohorts have been conducted in diabetic patients. One of the first studies conducted by Poblete Aro et al. [44] compared the effects of HIIT versus MICT on oxidative stress markers in adults with T2DM. Both exercise modalities improved general fitness and lipid profiles, but HIIT demonstrated superior effects on oxidative stress modulation. Specifically, the HIIT group experienced a significant reduction in MDA levels and a significant increase in GPx activity compared to both the MICT and control groups. No changes in SOD levels were observed in either group. Additionally, HIIT led to a marked increase in NO levels, suggesting enhanced endothelial function. Mortensen et al. [45] compared the effects of 11 weeks of MICT and low-volume HIIT on capillary ultrastructure and redox-related markers in individuals with T2DM, with skeletal muscle biopsies taken pre- and post-intervention. The study found no changes in levels of skeletal muscle SOD, NOX, or vascular endothelial growth factor in either group, suggesting limited impact on muscular oxidative stress or angiogenesis pathways. However, HIIT uniquely increased eNOS expression, indicating a potential benefit for endothelial function and NO-mediated vascular regulation. By contrast, the MICT group showed structural improvements in capillarization, reduced basement membrane thickness, and increased endothelium thickness. These changes were not observed with HIIT, indicating that MICT may be superior for reversing diabetes-related capillary remodeling. 

Sabouri et al. [46] investigated the effects of three different 12-week exercise interventions—HIIT, ST, and combined HIIT+ST—in T2DM patients. All three exercise interventions significantly improved antioxidant defenses, as evidenced by increased activities of SOD and GPx and total antioxidant capacity (TAC) compared to controls. Additionally, levels of inflammatory cytokines, including interleukin-6 (IL-6), C-reactive protein (CRP), and tumor necrosis factor (TNF-α), were significantly reduced in all training groups. Improvements in lipid profiles and glycemic parameters (e.g., blood glucose and insulin sensitivity) were also reported, with HIIT and HIIT+ST showing particularly favorable effects on aerobic fitness (VO_2peak_) and total cholesterol. Overall, the findings suggested that HIIT alone or in combination with ST is effective in improving redox balance, inflammatory status, and metabolic health in patients with T2DM, with combination training offering potentially greater cardiometabolic benefits. 

More recently, Al-Rafaw et al. [47] examined the effects of a 12-week HIIT program in male T2DM patients and found a significant increase in TAC, along with a significant decrease in 8-OHdG levels. Additionally, a reduction in tumor suppressor/apoptosis-related protein (p53) levels and an increase in cytochrome c oxidase (COX) levels indicated enhanced mitochondrial biogenesis and potentially lower apoptotic signaling. These molecular and antioxidant changes were closely linked with improved glycemic parameters, including fasting glucose, glycosylated hemoglobin (HbA_1c_), homeostatic model assessment for insulin resistance (HOMA-IR), insulin, and C-peptide. The results suggested that HIIT not only mitigates oxidative stress in T2DM but may also positively regulate mitochondrial function and insulin resistance via anti-apoptotic and antioxidative mechanisms.

To date, only one study has examined the longer-term (1 year) effects of HIIT on oxidative stress and antioxidant capacity in individuals with T2DM. Mallard et al. [48] conducted a randomized trial with 36 participants who completed 12 weeks of supervised treadmill-based HIIT (4 × 4 min at 90–95% of HR_max_) or MICT (constantly at 70% of HR_max_), followed by 40 weeks of home-based training at the same intensity. Biomarkers of oxidative stress (F2-isoprostanes and protein carbonyls), antioxidants (TAC and glutathione peroxidase activity), and inflammation (IL-6, IL-8, IL-10, and TNF-α) were assessed at baseline, 12 weeks, and 1 year. There were no significant changes in any oxidative stress or inflammatory markers during the initial 12-week supervised training period in either group. However, over the 1-year period, a trend toward maintenance of TAC was observed in the HIIT group compared to the MICT group, suggesting that long-term HIIT may potentially better preserve antioxidant defenses in T2DM. Notably, males in the HIIT group showed a reduction in protein carbonyl levels from 12 weeks to 1 year, pointing to sex-specific oxidative adaptations to longer-term HIIT in T2DM.

From a mechanistic standpoint, recent research highlights the crucial role of the Nrf2 pathway in mediating the beneficial redox adaptations to exercise by activating endogenous antioxidant defenses and promoting cellular protection against oxidative stress. Regular (particularly higher-intensity) exercise stimulates Nrf2 activity, leading to an increase in the transcription of genes encoding antioxidant and detoxification enzymes that combat ROS produced during physical activity [49]. Additionally, Nrf2 activation has been shown to improve mitochondrial efficiency and glucose uptake [49], offering a plausible pathway by which enhanced endogenous antioxidant capacity restores redox balance and supports metabolic control in T2DM.

In line with this, Kazemi et al. [50] demonstrated that a 12-week cycle ergometer-based HIIT intervention led to significant increases in H_2_O_2_ and Nrf2 levels in male patients with T2DM compared to controls. Concurrently, antioxidant enzyme levels (particularly CAT) were significantly elevated in the HIIT group. Moreover, HIIT improved various metabolic parameters (e.g., 12 h fasting plasma glucose, HbA_1c_, and blood lipids). These findings suggested that although HIIT acutely raises oxidative stress, it also activates protective antioxidant pathways (likely via Nrf2 signaling) in T2DM patients.

However, improvements in clinical outcomes (e.g., HbA_1c_) following HIIT have also been found to occur independently of changes in oxidative stress biomarkers. A recent meta-analysis of 22 randomized controlled HIIT trials reported significant improvements in glucose and lipid metabolism markers in T2DM cohorts [51], and it has been demonstrated that several exercise-related mechanisms can contribute independently to improved glycemic control, including enhanced glucose transporter type 4 (GLUT-4) translocation [52], increased muscle capillarization [53], and improved hepatic insulin signalling [52]. Consequently, endogenous antioxidant adaptation should be viewed as a complementary rather than singularly causal factor in the metabolic benefits conferred by HIIT. Additionally, it is important to note that T2DM is a heterogeneous disorder in which HIIT-related adaptations may vary with disease duration, pharmacotherapy (particularly insulin use), and habitual physical activity. It has been reported, for example, that shorter disease duration (<5 years) and an age of <60 years predict greater glucose and lipid metabolism improvements in T2DM patients following HIIT interventions, whereas long-standing disease cases and older patients (>60 years) exhibit attenuated responses [51].

Beyond research in T2DM cohorts, promising results have also been observed in populations with prediabetes and type 1 diabetes mellitus (T1DM). Bartlett et al. [54] investigated the effects of 10 weeks of low-volume HIIT on immune and metabolic functions in older adults (mean age: 71 years) with prediabetes. The intervention led to significant improvements in insulin sensitivity and cardiorespiratory fitness, accompanied by enhanced neutrophil functions. Before the training intervention, neutrophil chemotaxis, phagocytosis, and stimulated ROS production were significantly impaired, while basal ROS levels were elevated. Post-intervention, neutrophil chemotaxis and phagocytosis improved, stimulated ROS production increased, and basal ROS levels decreased, with levels approaching those of a young, healthy control group. Additionally, mitochondrial function in neutrophils—initially impaired—significantly improved in five participants. These findings suggest that in prediabetic older adults, HIIT may reverse immune cell oxidative stress dysfunction and improve neutrophil bioenergetics, potentially lowering infection risk and reducing disease progression. 

Farinha et al. [55] assessed the impact of 10 weeks of HIIT, ST, or a combination (HIIT+ST) in adults with T1DM. The study found that all training types led to improvements in antioxidant enzyme activities, including SOD, CAT, and TAC, and in glycemic control (HbA_1c_ and 8 h fasting glucose). However, levels of oxidative stress markers (TBARS, 8-OHdG, and oxidized LDL) and inflammatory cytokines (CRP, TNF-α, and IL-10) remained unchanged. Notably, the combination group (HIIT+ST) was the only one to show a reduction in daily insulin requirements, suggesting added benefits in insulin demand reduction when HIIT is combined with ST. In another study, Boff et al. [56] compared an 8-week cycle-ergometer-based HIIT (50–85% of HR_max_) and MICT (50% of HR_max_)—both 40 min per session—in young adults with T1DM. Interestingly, only the HIIT group demonstrated significant improvements in endothelial function (as assessed by flow-mediated dilation, FMD) and VO_2peak_, while the MICT group showed no significant change. However, levels of oxidative stress markers, including total thiol group concentrations and TBARS, as well as glycemic control and smooth muscle function (endothelium-independent vasodilation, measured via nitroglycerin-mediated dilation) remained unchanged across both exercise groups.

#### 4.2.2. Cardiovascular Disease

Sarvasti et al. [57] compared the mechanisms of cardiovascular protection between a two-week MICT and HIIT intervention in patients with stable coronary heart disease following coronary stenting. Eleven patients participated in this crossover design study, where each participant completed both exercise interventions with a two-week detraining period in between to eliminate any residual effects from the first intervention. Cardiovascular and endothelial function biomarkers were assessed before and after each intervention, including plasma levels of adrenaline, noradrenaline, eNOS, extracellular SOD (EC-SOD), and FMD. The results showed that HIIT significantly increased noradrenaline and eNOS concentrations compared to MICT and also better preserved EC-SOD activity and FMD. Mechanistically, noradrenaline was reported to have a direct and significant effect on both eNOS and FMD following HIIT. By contrast, under the MICT condition, noradrenaline increased eNOS levels, and EC-SOD activity mediated improvements in FMD. However, MICT was associated with reductions in EC-SOD activity and FMD, indicating a less favorable vascular and antioxidant response. Based on these findings, the authors concluded that HIIT was superior to MICT in enhancing cardiovascular protection in post-stenting coronary heart patients. 

Another more recent trial by Aispuru-Lanche et al. [58] explored the effects of different volumes of HIIT on endothelial function and oxidative stress in patients recovering from acute myocardial infarction. Patients were randomized into either a low-volume HIIT group (LV-HIIT, <10 min at high intensity), a high-volume HIIT group (HV-HIIT, >10 min at high intensity), or a control group. The study found that both LV-HIIT and HV-HIIT significantly improved vascular endothelial function (measured via FMD), with the HV-HIIT group showing a more pronounced effect compared to LV-HIIT. In parallel, both exercise groups experienced a reduction in carotid intima-media thickness and a decrease in oxidized LDL levels, while no significant changes were observed in the control group. Interestingly, oxidized LDL levels were inversely correlated with FMD improvements in both exercise groups. These findings suggested that even a low-volume HIIT protocol can elicit clinically meaningful improvements in oxidative stress and vascular health in post-myocardial infarction patients.

#### 4.2.3. Obesity

Henke et al. [59] investigated both the acute and chronic effects of HIIT on inflammatory cytokines and oxidative stress markers in sedentary postmenopausal obese women. Participants underwent a 4-week HIIT protocol consisting of two weekly sessions. Following the first HIIT session, acute increases were observed in levels of oxidative stress indictors (TBARS and advanced oxidation protein products) and inflammation markers, including IL-6, IL-10, and monocyte chemoattractant protein-1 (MCP-1). However, after completing 4 weeks of training, resting levels of pro-inflammatory IL-6 decreased, while levels of anti-inflammatory IL-1ra and IL-10 increased, indicating a shift toward an anti-inflammatory profile. In the final HIIT session, only levels of IL-6, IL-10, and IL-1ra were elevated post-exercise, with no changes in MCP-1, IL-1β, or TBARS levels, suggesting reduced acute inflammatory and oxidative responses. Overall, the study indicated that short-term HIIT promotes anti-inflammatory adaptations in obese postmenopausal women, with only small changes in oxidative stress markers over time.

De Matos et al. [60] investigated the effects of 8 weeks of HIIT on HOMA-IR and skeletal muscle oxidative metabolism in physically inactive individuals with obesity, with a particular focus on comparing obese insulin-resistant (OBR) and obese non-insulin-resistant (OB) phenotypes. HIIT led to an increase in VO_2peak_ in both OB and OBR, but without significant changes in total body fat. A decrease in HOMA-IR was observed in the OBR group, indicating improved insulin sensitivity. On the muscular level, both groups exhibited increased phosphorylation of insulin signaling proteins (IRS Tyr612 and Akt Ser473) and increased content of beta-hydroxyacid dehydrogenase (β-HAD) and COX-IV, markers associated with oxidative metabolism. Additionally, a decrease in ERK1/2 phosphorylation was noted in the OBR group, suggesting a modulation of pathways related to inflammation or cell stress. Overall, the study demonstrated that 8 weeks of HIIT can improve insulin signaling and enhance skeletal muscle oxidative capacity in individuals with obesity, particularly benefiting those with baseline insulin resistance.

More recently, Flensted-Jensen et al. [61] investigated the effects of a 6-week cycle ergometer-based low-volume HIIT protocol on oxidative stress markers and antioxidant capacity in sedentary, obese individuals at risk of developing T2DM. Following the intervention, there was a ~60% reduction in muscle H_2_O_2_ emission, indicating reduced ROS production. In parallel, there was a ~35% increase in manganese superoxide dismutase (MnSOD) protein levels and a strong (non-significant) trend toward increased CAT levels (~73%). Additionally, mitochondrial respiratory capacity improved by 19%, VO_2peak_ increased by ~7%, and body fat decreased by 1.7%. These findings indicated that a low-volume HIIT protocol can significantly reduce oxidative stress while enhancing antioxidant defenses and mitochondrial function in obese individuals at increased metabolic risk.

#### 4.2.4. Hypertension

Gunnarsson et al. [62] explored the effects of a HIIT-type exercise program on vascular responsiveness, blood pressure, and skeletal muscle oxidative status in postmenopausal women, specifically comparing hypertensive (HYP) and normotensive (NORM) women. The 10-week intervention consisted of biweekly small-sided floorball sessions involving repeated 3–5-minute high-intensity intervals interspersed with 1–3 minutes of rest. At baseline, hypertensive women exhibited reduced NO-mediated vasodilator responsiveness, as evidenced by a significantly lower increase in leg vascular conductance (LVC) in response to sodium nitroprusside (SNP) infusion. Following the exercise intervention, acetylcholine- and SNP-induced LVC significantly improved in the HYP group, indicating restoration of endothelial and smooth muscle responsiveness to NO. In addition to vascular improvements, the training program led to a marked reduction in blood pressure. Systolic and diastolic blood pressure decreased by 15 mmHg and 9 mmHg, respectively, in the HYP group, while the NORM group experienced a 10 mmHg reduction in systolic pressure. Regarding oxidative stress and antioxidant responses, HIIT did not alter the skeletal muscle content of eNOS or antioxidant enzymes, including SOD1, SOD2, GPX, CAT, or NOX, in either group. However, in hypertensive women, an increase in dynamin-related protein 1 content was observed, suggesting enhanced mitochondrial fission potential. The findings suggested that while HIIT improves vascular function and blood pressure in hypertensive postmenopausal women, it may not significantly alter levels of markers of oxidative stress or NO-bioavailability in skeletal muscle, particularly in women >6 years post-menopause. 

#### 4.2.5. Non-Alcoholic Fatty Liver Disease (NAFLD)

Guo et al. [63] investigated the effects of a 12-week supervised HIIT program compared to MICT in patients with NAFLD. Both exercise interventions significantly reduced oxidative stress markers, including MDA and protein carbonyl levels, suggesting a beneficial effect on systemic redox balance. However, these improvements were more pronounced in the HIIT group. In addition to redox changes, HIIT also led to greater improvements in cardiorespiratory fitness, muscular strength, liver enzyme levels (ALT and AST), inflammatory markers (CRP and IL-6), lipid profile, and insulin sensitivity (HOMA-IR), highlighting HIIT as a viable exercise strategy in managing multiple aspects of metabolic health in NAFLD.

**Table 1 antioxidants-14-00937-t001:** Interventional studies examining the effects of HIIT on oxidative stress status in cardiometabolic disease cohorts.

Ref.	Design	Condition	*n*	HIIT Protocol	Duration (Weeks)	Main effects on Redox Outcomes	Main Effects on Other Clinical Outcomes
[44]	RCT	T2DM	14	Treadmill; 4–6 × 1 min @ 80–85% VO_2peak_; 4 min recovery @ 50–60% VO_2peak_; total session time: 30–40 min; 3×/week	12	- ↓MDA, and↑GPx - ↑NO - No change in SOD	- Improved lipid profiles - Improved VO_2max_
[45]	RCT	T2DM	11	Cycle ergometer; 10 × 1 min @ 95% W_peak_; 1 min recovery @ 20% W_peak_; total session time: 20 min; 3×/week	11	- No change in SOD2, and NOX - ↑ eNOS expression in skeletal muscle	- No structural vascular adaptations - No change in VEGF
[46]	RCT	T2DM	30 (15/15)	Group-1 (*n* = 15): cycle ergometer; 10 × 1 min @ 85–90% HR_max_; 1 min recovery; total session time: 30 min; 3×/week Group-2 (*n* = 15): HIIT+ ST (7 machine-based strength exercises, 3 sets × 8 repetitions); total session time: 70 min; 3×/week	12	- ↑SOD and↑GPX in both groups - ↑TAC in both groups	- ↓IL-6, CRP, and TNF-αin both groups - ↓total cholesterol in both groups - ↑VO_2peak_ in both groups
[47]	Quasi- Experimental Study	T2DM	30	Treadmill; 4 × 4 min @ 80–85% HR_max_; 3 min recovery @ 70% HR_max_; total session time: 40 min; 3×/week	12	- ↑TAC, and↓8-OHdG - ↓p53 protein - ↑COX, and↑mitochondrial DNA content in muscle tissue	- ↓Fasting glucose, HbA_1c_, HOMA-IR, fasting insulin, and C-peptide
[48]	RT	T2DM	20	Treadmill; 4 × 4 min @ 90–95% HR_max_; 3 min recovery @ 70% HR_max_; total session time: 40 min; 3×/week	12, and 48 (follow-up)	- No effects on redox markers - ↓TAC - Sex-specific findings: ↓protein carbonyls (12 weeks to 1 year) only in males	- no changes in inflammation
[50]	RCT	T2DM	16	Cycle ergometer; 10 × 1 min @ 90% HR_max_; 1 min recovery @ 50 W; total session time: 25 min; 3×/week	12	- ↑H_2_O_2_, and Nrf2 - ↑CAT	- ↓12 h fasting and postprandial glucose, HbA_1c_, total cholesterol, triglycerides
[54]	Exploratory Study	Pre- diabetes	10	Treadmill; 10 × 1 min @ 80–90% HRR; 3 min recovery @ 50–60% HRR; total session time: 30 min; 3×/week	10	- ↓Basal ROS - ↑Neutrophil chemotaxis, phagocytosis, stimulated ROS - ↑Neutrophil mitochondrial function	- ↓Overnight fasting glucose and insulin - ↑VO_2peak_
[55]	RCT	T1DM	19 (9/10)	Group-1 (*n* = 9): cycle ergometer; 10 × 1 min @ 90% HR_max_; 1 min active recovery; total session time: 25 min; 3×/week Group-2 (*n* = 10): HIIT+ST (7 machine-based strength exercises (3 sets × 8 repetitions); total session time: 65–70 min; 3×/week	10	- ↑TAC, catalase, SOD in both groups - No change in oxidative stress markers in both groups	- ↓8 h fasting glucose and HbA_1c_ in both groups - No change in inflammation in both groups - ↓Daily insulin dosage, only in HIIT plus ST - ↓Daily insulin dosage, only in HIIT plus ST
[56]	RCT	T1DM	9	Cycle ergometer; 3–6 × 1 min @ 80–85% HR_max_; 4 min recovery @ 50% HR_max_; total session time: 25–40 min; 3×/week	8	- No change in oxidative stress markers	- ↑VO_2peak_ - ↑Endothelial function - No change in glycemic control and smooth muscle function
[57]	Crossover- experimental	Stable coronary artery disease	11	Treadmill; 4 × 4 min @ 60–80% HRR; 3 min recovery @ 40–50% HRR; total session time: 30 min; 3×/week	2	- ↑EC-SOD activity - ↑eNOS levels	- ↑endothelial function
[58]	RCT	Post- myocardial infarction	56 (28/28)	LV-HIIT (*n* = 28): mixed treadmill (4 min intervals @ VT2 to HR_max_; 3 min active recovery) and cycle ergometer (30 s intervals @ VT2 to HR_max_; 1 min active recovery); total session time: 20 min; 2×/week HV-HIIT (*n* = 28): similar protocol, but gradually increasing volume; total session time: 20–40 min; 2×/week	16	- ↓oxidized LDL in both groups	- ↑endothelial function in both groups - ↓vascular wall thickness in both groups
[59]	Quasi- Experimental Study	Obesity	10	Cycle ergometer; 10 × 1 min @ 85% HR_max_; 0.75 s recovery @ 40% HR_max_; total session time: 31.5 min; 2×/week	4	- Acute post-session increases in TBARS and AOPP levels improved after 4 weeks - No chronic changes in resting oxidative stress markers	- ↓resting IL-6 - ↑resting IL-10, and IL-Ra
[60]	Quasi- Experimental Study	Obesity with (OBR) or without (OB) insulin resistance	17 (9/8)	Cycle ergometer; 8–12 × 1 min @ 80–110% W_peak_; 1 min recovery @ 30 W; total session time: 16–24 min; 3×/week	8	- ↑β-HAD, and COX-IV content in both groups - ↓ERK1/2 phosphorylation only in OBR - ↑IRS Tyr612 phosphorylation, and Akt Ser473 phosphorylation in both groups	- ↓HOMA-IR only in OBR - ↑VO_2peak_ in both groups - No change in body fat in both groups
[61]	Exploratory Study	Obesity	12	Cycle ergometer; 5 × 1 min @ 125% VO_2peak_, 90 s recovery; total session time: 15 min; 3×/week	10	- ↓H_2_O_2_, and MnSOD - trend for↑CAT - ↑mitochondrial capacity	- ↑VO_2peak_ - ↓body fat
[62]	Quasi- Experimental Study	Postmenopausal women with (HYP) and without (NORM) hyper- tension	17 (9/8)	Intermittent small-sided floorball games; 4–5 × 3–5 min @ >85% HR_max_; 3–1 min active recovery; total session time: 40–60 min; 2×/week	10	- No change in skeletal muscle content of SOD1, SOD2, GPX, CAT, NOX - No change in eNOS - ↑DRP1 increased only in HYP - ↑ACh- and SNP-induced LVC improved only in HY	- ↓Systolic/diastolic BP reduced in HYP ↓Systolic BP reduced in NORM
[63]	RCT	NAFLD	10	Treadmill; 4 × 4 min @ 85–95% HR_peak_, 3 min active recovery; total session time: 30–40 min session; 3×/week	12	- ↓MDA and protein carbonyls	- ↑CRF and muscular strength - ↓Liver enzymes (ALT, AST) - ↓Inflammation (CRP, IL-6) - ↓LDL, triglycerides, HOMA-IR

Abbreviations. RCT: randomized-controlled trial; T2DM: type 2 diabetes mellitus; VO_2peak_: peak oxygen uptake; MDA: malondialdehyde; GPx: glutathione peroxidase; NO: nitric oxide; SOD: superoxide dismutase; VO_2max_: maximal oxygen uptake; W_peak_: peak power output; NOX: nicotinamide adenine dinucleotide phosphate oxidase; eNOS: endothelial nitric oxide synthase; VEGF: vascular endothelial growth factor; HR_max_: maximal heart rate; HIIT: high-intensity interval training; ST: strength training; TAC: total antioxidant capacity; IL: interleukin; CRP: C-reactive protein; TNF-α: tumor necrosis factor; 8-OHdG: 8-hydroxy-2′-deoxyguanosine; p53: tumor suppressor/apoptosis-related protein; COX: cytochrome c oxidase; HbA_1c_: glycosylated hemoglobin; HOMA-IR: homeostatic model assessment for insulin resistance; H_2_O_2_: hydrogen peroxide; Nrf2: nuclear factor erythroid 2–related factor 2; CAT: catalase; HRR: heart rate reserve; ROS: reactive oxygen species; T1DM: type 1 diabetes mellitus; EC-SOD: extracellular SOD; LV: low-volume; HV: high-volume; VT2: 2nd ventilatory threshold; TBARS: thiobarbituric acid reactive substances; AOPP: advanced oxidation protein products; β-HAD: beta-hydroxyacid dehydrogenase; MnSOD: manganese superoxide dismutase; DRP-1: dynamin-related protein1; SNP: sodium nitroprusside; ACh: acetylcholine; LVC: leg vascular conductance; BP: blood pressure; CRF: cardiorespiratory fitness.

## 5. Conclusions and Perspectives

Across diverse cardiometabolic disease populations, including obese, T2DM, and CVD patients, HIIT has emerged as a promising non-pharmacological intervention to improve redox balance. While the magnitude and consistency of oxidative stress modulation varied between cohorts, the collective evidence indicates that HIIT can reduce oxidative stress and support antioxidant capacity mechanisms in these populations. The most consistently reported findings are reductions in systemic markers of oxidative stress, such as oxidized LDL, and upregulation of key antioxidant enzymes like SOD. These improvements appear to be closely linked with enhanced endothelial function and may contribute to the broader vascular and metabolic benefits of HIIT. Notably, HIIT induced beneficial adaptations even in the absence of changes in body composition, suggesting that the improvements in redox status are driven more by physiological stress and mitochondrial signaling than by weight loss per se. 

However, these findings should be considered in the context of a still heterogeneous and evolving body of evidence. Many conclusions were derived from small-to-moderate-sized randomized controlled or quasi-experimental studies that varied widely in design, intervention protocols, study duration, and biomarker selection. Moreover, the mechanistic contribution of oxidative stress modulation to clinical improvements remains context-dependent. In some studies, changes in levels of redox biomarkers occurred alongside metabolic and vascular benefits, while in others, clinical gains were also evident in the absence of measurable oxidative adaptations. This suggests that redox modulation may serve a complementary rather than singularly causal role in the overall efficacy of HIIT.

Importantly, age, hormonal status, and individual redox responsiveness appear to influence outcomes. For instance, in older adults and postmenopausal women, vascular improvements have been observed without significant alterations in levels of skeletal muscle oxidative biomarkers, potentially reflecting diminished adaptive redox plasticity. Nonetheless, these populations still exhibit clinically meaningful benefits in cardiometabolic indices such as decreased blood pressure, indicating that HIIT can confer value despite an unchanged redox status. While some studies suggest that higher-volume HIIT protocols may yield greater redox benefits, lower-volume regimens have also been shown to induce protective changes. This suggests that while dose–response relationships appear to exist, the threshold for benefits is relatively low, reinforcing the accessibility of HIIT as a therapeutic modality. 

When comparing HIIT to more traditional MICT prescriptions, research has consistently shown that both modalities can produce significant reductions in body mass and body composition, while HIIT appears to offer greater improvements in VO_2max_ and some cardiometabolic risk indices [9,27,28,64]. Importantly, improved VO_2max_ (the key marker of cardiorespiratory fitness) is strongly associated with reduced mortality risk [65], and mitochondrial enzyme activities (e.g., citrate synthase and complex I) have been reported to increase more after HIIT compared to MICT, demonstrating more pronounced mitochondrial adaptation [66]. However, given a dose–response association between the amount of physical activity and weight loss [67], higher-volume MICT may be more effective for sustainable body weight reduction. Thus, exercise programs that integrate both HIIT and MICT may offer synergistic benefits (e.g., HIIT to rapidly enhance cardiorespiratory fitness and mitochondrial capacity in a more time-efficient and feasible way, and MICT to support sustained weight management). This blended strategy deserves further exploration in future clinical trials.

Taken together, the available evidence supports the role of HIIT as a time-efficient, redox-modifying strategy capable of improving cardiometabolic health across a range of clinical populations. Nevertheless, greater methodological consistency is needed to clarify the dose–response relationships and population-specific effects underlying these adaptations. 

Looking ahead, several avenues warrant further exploration to better harness HIIT as a redox-modifying therapy. First, given the mixture of study designs and study populations, future studies should aim to investigate interindividual variability by stratifying for age, duration and severity of disease, sex hormone status, habitual physical activity patterns, dietary intake, and concurrent pharmacotherapy. Second, there is a need to explore the interaction between training intensity, volume, and program duration in determining redox outcomes. Third, longer-lasting studies are needed to assess the long-term sustainability of redox improvements and their predictive value for clinical endpoints such as cardiovascular events and disease progression. For instance, while there is emerging evidence that certain cardiometabolic adaptations following HIIT may persist beyond the active training period [68,69,70], further research is warranted to determine whether redox adaptations show similar persistence during periods of detraining. Finally, the potential synergistic effects of combining HIIT with adjunctive strategies, such as targeted nutrition or pharmacological agents, merit deeper exploration. As the body of research on HIIT and redox biology in cardiometabolic diseases continues to grow, further well-designed, hypothesis-driven studies and future systematic reviews will be essential to quantitatively assess intervention effects and guide clinical translation.

## 6. Practical Recommendations and Safety Considerations

Protocols using 4 × 4 min treadmill intervals or 10 × 1 min cycle ergometer intervals at 85–90% of maximal heart rate, interspersed with active recovery, have been effective and feasible in diverse cardiometabolic cohorts. However, low-volume HIIT protocols (<10 minutes at high intensity) have also shown promising benefits to stimulate improvements in endothelial function and reductions in levels of oxidative stress markers, suggesting that volume can be flexibly adapted based on health status and exercise tolerance. Such low-volume protocols could serve as a preparatory strategy to build foundational fitness and tolerance before progressing to more demanding regimens, particularly for patients who are physically deconditioned, older, or otherwise unable/unwilling to engage in higher-volume protocols. Both structured HIIT (ergometer-based) and more playful formats like small-sided games (e.g., floorball) have been shown to be effective, allowing for personalized, enjoyable, and sustainable regimens.

Regardless of the specific format, HIIT should be introduced progressively, with gradual increases in intensity and volume over time. Initial training loads should be carefully determined, especially in deconditioned or higher-risk individuals, and closely monitored for tolerance and recovery. It is advisable to incorporate 2 to 3 sessions per week, allowing for sufficient recovery and adaptation.

Finally, it is important to highlight that while HIIT is generally well tolerated, especially under supervised conditions, individuals with cardiometabolic disorders should undergo medical evaluation and risk stratification prior to engaging in high-intensity exercise. Where available, a baseline cardiopulmonary exercise test can help to determine safe heart rate zones and detect any contraindications to vigorous effort. Given the higher physiological demands of HIIT, programs should ideally be supervised by qualified health professionals, such as exercise physiologists or cardiac rehabilitation staff, particularly in populations with advanced disease, recent acute coronary events, or uncontrolled comorbidities. Moreover, HIIT protocols should be individualized, tailored to a patient’s age, comorbidities, medication profile, and baseline fitness. In some cases, modified HIIT formats with lower peak intensities or longer recovery periods may be necessary to ensure safety without compromising efficacy. Importantly, proper familiarization periods, as used in several studies, can be helpful in adjusting to higher workloads.

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
