# Peer review of "High-Intensity Interval Training as Redox Medicine: Targeting Oxidative Stress and Antioxidant Adaptations in Cardiometabolic Disease Cohorts"

_antioxidants, 2025, doi:10.3390/antiox14080937_

Round 1

Reviewer 1 Report

The paper reviews the utility of 'high-intensity interval training (HIIT' as a means of significantly improving cardiometabolic health. Having defined oxidative stress at the physiological and molecular level, the review goes on to discuss the role that physical exercise can play in preventing and managing cardiometabolic disorders (including: type 2 diabetes mellitus (T2DM), cardiovascular disease and metabolic syndrome). While oxidative stress can act as a driver of disease progression, it was interesting to read that vigorous-intensity exercise can cause an increase in the generation of reactive oxygen species (ROS), and therefore a transient state of oxidative stress. HIIT however may limit the duration of oxidative exposure, cause a surge in ROS levels, but this in turn triggers the activation of signalling pathways that enhance the synthesis of endogenous antioxidants, so that over time consistent exposure to oxidative challenge can pave the way for measurable adaptations to the body's endogenous antioxidant systems. For compromised sufferers with significant cardiometabolic disorders who are 'time-poor', a carefully prescribed HIIT regimen can be used as a possible therapeutic modality to alleviate oxidative stress in a way that can bring long-lasting benefits.

The author should be commended for the precision of the writing all the way through the manuscript. The narrative was straightforward to read, and did not contain any ambiguous terminology to detract from what are compelling arguments for the use of HIIT in a treatment protocol. One minor criticism is the lack of diagrammatic representations that could be used to illustrate how oxidative stress is manifested at the cellular level, and the signalling systems that are used to restore equilibrium when ROS have elevated to potentially damaging thresholds. The relationship between HIIT and oxidative stress in clinical populations is interesting and the author was right to point out the need for better understanding of the progression of acute responses to chronic adaptations before HIIT can be carefully implemented to address cardiometabolic care. The obesity linked to T2DM conundrum is a cause for concern, and the review highlights some important studies with HIIT that have been carried out on diabetic patients. Some of the exercise interventions (with HIIT and its variations) are quite long and further reading of the references cited may show how the designers of these 'trials' were able to solicit the commitment from these patients for these quite long durations. The authors managed to find reference to a '1-year' trial with HIIT. This presumably generated a range of interesting results, summarised by the observation that males showed a reduction in protein carbonyls. Table 1 gives a very good overview of the range of interventional studies examining the effects of HIIT on oxidative stress in patient groups, and nearly all seem to feature metabolic markers that need to be militated against to improve cardiometabolic health.

Author Response

Major comments

The paper reviews the utility of 'high-intensity interval training (HIIT' as a means of significantly improving cardiometabolic health. Having defined oxidative stress at the physiological and molecular level, the review goes on to discuss the role that physical exercise can play in preventing and managing cardiometabolic disorders (including: type 2 diabetes mellitus (T2DM), cardiovascular disease and metabolic syndrome). While oxidative stress can act as a driver of disease progression, it was interesting to read that vigorous-intensity exercise can cause an increase in the generation of reactive oxygen species (ROS), and therefore a transient state of oxidative stress. HIIT however may limit the duration of oxidative exposure, cause a surge in ROS levels, but this in turn triggers the activation of signalling pathways that enhance the synthesis of endogenous antioxidants, so that over time consistent exposure to oxidative challenge can pave the way for measurable adaptations to the body's endogenous antioxidant systems. For compromised sufferers with significant cardiometabolic disorders who are 'time-poor', a carefully prescribed HIIT regimen can be used as a possible therapeutic modality to alleviate oxidative stress in a way that can bring long-lasting benefits.

Detailed comments

The author should be commended for the precision of the writing all the way through the manuscript. The narrative was straightforward to read, and did not contain any ambiguous terminology to detract from what are compelling arguments for the use of HIIT in a treatment protocol. One minor criticism is the lack of diagrammatic representations that could be used to illustrate how oxidative stress is manifested at the cellular level, and the signalling systems that are used to restore Equilibrium when ROS have elevated to potentially damaging thresholds. The relationship between HIIT and oxidative stress in clinical populations is interesting and the author was right to point out the need for better understanding of the progression of acute responses to chronic adaptations before HIIT can be carefully implemented to address cardiometabolic care. The obesity linked to T2DM conundrum is a cause for concern, and the review highlights some important studies with HIIT that have been carried out on diabetic patients. Some of the exercise interventions (with HIIT and its variations) are quite long and further reading of the references cited may show how the designers of these 'trials' were able to solicit the commitment from these patients for these quite long durations. The authors managed to find reference to a '1-year' trial with HIIT. This presumably generated a range of interesting results, summarised by the observation that males showed a reduction in protein carbonyls. Table 1 gives a very good overview of the range of interventional studies examining the effects of HIIT on oxidative stress in patient groups, and nearly all seem to feature metabolic markers that need to be militated against to improve cardiometabolic health.

Response to Reviewer #1

First of all, I would like to extend my sincere gratitude to Reviewer #1 for the thoughtful and highly encouraging feedback on my manuscript. I am pleased to know that the clarity, structure, and scientific relevance of the narrative review were well received. Your kind words regarding the precision of writing and the compelling discussion are truly appreciated.

Comment:
“One minor criticism is the lack of diagrammatic representations that could be used to illustrate how oxidative stress is manifested at the cellular level, and the signalling systems that are used to restore equilibrium when ROS have elevated to potentially damaging thresholds.”

Response:

I appreciate this valuable suggestion. Indeed, a graphical representation could aid in the conceptual understanding of complex cellular processes but I deliberately chose not to include such illustrations in the current manuscript for several interrelated reasons explained in detail below.

1. Scope and focus of the review:

The manuscript is a narrative review with a clinical and translational orientation, designed to integrate current findings on the interaction between HIIT and oxidative stress in cardiometabolic diseases. While the cellular mechanisms of oxidative stress are critical background knowledge (and are therefore discussed in introductory sections), these have been extensively detailed in prior foundational reviews. As such, I opted to direct the reader to these existing comprehensive resources rather than replicate them in graphical form, which might have been redundant.

2. Narrative flow and clarity:

I aimed for a concise and fluent reading experience, prioritizing clarity and depth in textual explanation over visual elements. All molecular mechanisms relevant to the specific arguments were discussed within the text, and I aimed to structure my narrative to build understanding progressively, minimizing the need for auxiliary visual aids.

3. Space constraints:

The inclusion of additional figures would have required more space to be meaningful and accurate, potentially disrupting the balance between textual synthesis and visual content. As the manuscript already includes a comprehensive Table 1, summarizing intervention studies and metabolic markers affected by HIIT, I felt this provided a clear and digestible reference point for readers without additional visual burden.

That explained, I genuinely value your important feedback and I have now included a sentence in the Introduction Section to explicitly direct readers to recommended reviews that offer detailed visual representations of redox biology and oxidative signaling pathways in metabolic disease. I believe this approach respects the reader's interest while preserving the focused scope of the present review.

A sentence has been added to the Section 3.3 (page 3, lines 118-121):

„Readers seeking a more detailed discussion and visual representations of the molecular mechanisms underlying oxidative stress at the cellular level in cardiometabolic diseases are referred to previous comprehensive reviews [1,2].“

I hope this rationale satisfactorily addresses your concern and underscores my approach to manuscript structure and content. However, should you and/or the journal feel that the inclusion of a targeted figure would add significant value, I am, of course, more than willing to revisit this and work on a suitable addition.

Reviewer 2 Report

The article "High-Intensity Interval Training as Redox Medicine: Targeting Oxidative Stress and Antioxidant Adaptations in Cardiometabolic Disease Cohorts" presents an interesting topic, but it requires a critical analysis of the information. It would be more suitable as a systematic review, including a PICO question and systematization of the evidence.

The introduction lacks a logical sequence, failing to clearly outline knowledge gaps, the link between HIIT and oxidative stress, and the need for a redox medicine perspective.

Section 2 is very limited, providing a general and uncritical overview of oxidative stress's role in cardiometabolic disease pathophysiology. A critical analysis is needed to determine whether oxidative stress is a cause or effect, and the impact of antioxidant therapy or endogenous antioxidant system precursors on disease progression.

When discussing HIIT and cardiometabolic diseases, factors such as age, previous physical activity/sedentary lifestyle, supplementation, HIIT protocol, and effects on oxidative stress biomarkers in populations must be considered.

The analysis of cardiometabolic diseases presents an extensive summary of evidence without critical analysis, prioritizing quantity over quality. For example, type 2 diabetes is a heterogeneous disease, and HIIT's effects may vary depending on factors like disease duration, insulin use, and sedentary lifestyle. What is the theoretical basis for suggesting that increased endogenous antioxidant capacity improves redox balance in these patients? What do multivariate analyses indicate? Could HIIT improve parameters like HbA1c independently of oxidative stress markers?

The table should separate the main effects column into 2 columns: one indicating markers of oxidative stress, and the other one describing the parameters of disease improvement (glycemic control, blood pressure, blood lipids).

The conclusion lacks support due to the mixture of study designs and lack of systematization.

Author Response

Major comments

The article "High-Intensity Interval Training as Redox Medicine: Targeting Oxidative Stress and Antioxidant Adaptations in Cardiometabolic Disease Cohorts" presents an interesting topic, but it requires a critical analysis of the information. It would be more suitable as a systematic review, including a PICO question and systematization of the evidence.

Response

Dear reviewer,

First of all, I would like to thank you very much your time and effort taken to review the manuscript and for your valuable comments and suggestions on how to improve the article. I would also thank you for your kind words of introduction and I am glad that you found the paper interesting to read.

Please find my point-by-point responses to your comments below. I have made revisions according to your comments and suggestions, wherever possible.

Regarding your major comment, I sincerely thank you for this valuable suggestion. The choice to pursue a narrative review format (instead of a systematic review) was made with careful consideration of the current state of the literature on this emerging topic. Below, I outline the specific rationale behind this decision:

1. Heterogeneity of study designs and populations

As highlighted in several places in the article, the existing body of literature examining the effects of HIIT on oxidative stress and antioxidant responses in cardiometabolic disease populations is still heterogeneous in terms of study design, participant characteristics (e.g., age, sex, disease type), HIIT protocols, outcome measures (e.g., biomarkers used), and follow-up durations. This diversity presents significant challenges to standardizing inclusion criteria and outcomes that are required for a rigorous systematic review with a PICO framework.

2. Limited number of RCTs

While interest in this topic is growing, the number of randomized controlled trials in cardiometabolic disease cohorts remains limited. Thus, in my opinion, it is still a little bit early for a rigorous systematic review at this point in time.

3. Purpose of the Review

The primary aim of my article is to provide a conceptual framework that integrates current findings, highlights important knowledge gaps, and contextualizes oxidative stress as a potential therapeutic target in exercise research. In my opinion, this approach is best suited to a narrative review, which allows for critical reflection, the identification of emerging topics, and the generation of hypotheses for future investigation, particularly valuable in fields where research is still evolving. Given the current translational but exploratory nature of HIIT as “redox medicine,” a narrative format enables to connect mechanistic insights with clinical implications across various subfields.

I appreciate your suggestion and fully agree that as the field matures, future systematic reviews and meta-analyses will be valuable for quantifying intervention effects and guiding clinical practice. Thus, in response to your comment, I have added a brief section to the Conclusions and Perspectives Section (page 14, lines 567-570):

„As the body of research on HIIT and redox biology in cardiometabolic diseases continues to grow, further well-designed, hypothesis-driven studies and future systematic reviews will be essential to quantitatively assess intervention effects and guide clinical translation.“

Detailed comments

#1

The introduction lacks a logical sequence, failing to clearly outline knowledge gaps, the link between HIIT and oxidative stress, and the need for a redox medicine perspective.

Response:

Thank you for this suggestion. Accordingly, I have revised the section at several place to better introduce the scope, rationale, and perspective of the review in a logical progression.

Please see: Introduction Section, page 2, lines 42-44, and lines 52-64:

„Among non-pharmacological interventions, lifestyle modifications, particularly regular physical exercise, are foundational in preventing and managing cardiometabolic diseases. Importantly, exercise functions as a physiological stressor that can induce […]“

„However, the impact of HIIT on oxidative stress regulation remains insufficiently understood. While some studies report beneficial antioxidant adaptations, others raise concerns regarding pro-oxidant responses, especially in individuals with compromised redox homeostasis [11–13]. Moreover, questions remain about the optimal dose, safety, and patient suitability of HIIT in disease contexts where oxidative stress is already elevated. In view of these uncertainties and knowledge gaps, it is timely to shed light on the conceptual framing of HIIT as a tool within redox medicine, and to critically discuss its potential to alter redox signaling pathways for therapeutic benefit. Therefore, this narrative review aims to (i) synthesize recent findings on HIIT-induced redox responses in cardiometabolic disease cohorts, (ii) identify gaps and controversies in the literature, (iii) address practical recommendations, and (iv) provide a forward-looking evaluation of the potential of HIIT as a non-pharmacological redox-modifying intervention within cardiometabolic disease settings.“

#2

Section 2 is very limited, providing a general and uncritical overview of oxidative stress's role in cardiometabolic disease pathophysiology. A critical analysis is needed to determine whether oxidative stress is a cause or effect, and the impact of antioxidant therapy or endogenous antioxidant system precursors on disease progression.

Response:

I thank you for this constructive feedback. I acknowledge that Section 2 provides a largely descriptive overview of oxidative stress in cardiometabolic disease, which may appear general in nature. The intention was to set the foundational background for the main focus of the review: The effects of HIIT on redox balance in clinical populations with cardiometabolic diseases. Given that this is a narrative review primarily focused on the intervention (HIIT) rather than a detailed analysis of redox pathophysiology or pharmacological antioxidant therapies, I aimed to provide a concise yet integrative introduction to oxidative stress as a shared mechanistic pathway across these conditions.

That explained, I agree, however, that a more critical reflection on the cause-effect relationship of oxidative stress and on the therapeutic implications of antioxidant strategies would enhance the quality of this section. According to your concern, I have therefore added clarifying statements to highlight that oxidative stress plays both causative and consequential roles in disease progression, and I included a brief discussion of antioxidant supplementation, contrasting it with the potential of exercise-induced endogenous antioxidant adaptation, which is the central theme of this review. Please see Section 2.2, page 3, lines 111-121:

„In this context, a critical and ongoing question is whether oxidative stress repre-sents a primary driver or a secondary consequence of disease pathology. The available evidence suggests a bidirectional relationship, whereby oxidative stress both arises from metabolic dysregulation and, in turn, exacerbates it through self-reinforcing mechanisms [3,17]. This dual role complicates the interpretation of redox biomarkers and presents a challenge for therapeutic targeting. As such, strategies aimed at restoring redox homeostasis must consider not only the suppression of excess reactive species but also the preservation of physiological ROS signaling required for cellular adaptation and metabolic regulation. Readers seeking a more detailed discussion and visual representations of the molecular mechanisms underlying oxidative stress at the cellular level in cardiometabolic diseases are referred to previous comprehensive reviews [1,2].“

and Section 2.3, page 3-4, lines 137-146:

„While exogenous antioxidant therapies, such as targeted supplementation with vitamins C and E, polyphenols, or synthetic scavengers, have shown some promise in preclinical models, their clinical efficacy in reducing cardiovascular or metabolic disease risk remains inconclusive. Systematic reviews have reported only minimal or no significant impact on cardiometabolic endpoints [21–23], and concerns have been raised regarding the potential blunting of beneficial ROS-mediated signaling cascades essential for metabolic and muscular adaptations [24]. These limitations have shifted attention toward interventions that enhance the body’s endogenous antioxidant capacity, rather than simply neutralizing ROS. In this context, physical exercise has emerged as a promising strategy for inducing redox adaptations through hormetic mechanisms.“

#3

When discussing HIIT and cardiometabolic diseases, factors such as age, previous physical activity/sedentary lifestyle, supplementation, HIIT protocol, and effects on oxidative stress biomarkers in populations must be considered.

Response:

I appreciate your insightful comment and I agree that multiple individual and methodological factors can significantly influence the redox responses to HIIT in clinical populations. I have already discussed some of these factors in the Conclusions and Perspectives section of the previous version of the manuscript. However, in response to your comment and in order to address these important aspects in more detail, I have expanded this discussion passage. Please see: page 13, lines 524-531:

„Importantly, age, hormonal status, and individual redox responsiveness appear to influence outcomes. For instance, in older adults and postmenopausal women, vascular improvements have been observed without significant alterations in skeletal muscle oxidative biomarkers, potentially reflecting diminished adaptive plasticity. Nonetheless, these populations still exhibit clinically meaningful benefits in cardiometabolic indices such as decreased blood pressure, indicating that HIIT can confer value despite an unchanged redox status. While some studies suggest that higher-volume HIIT protocols may yield greater redox benefits, lower-volume regimens have also been shown to induce protective changes.“

#4

The analysis of cardiometabolic diseases presents an extensive summary of evidence without critical analysis, prioritizing quantity over quality. For example, type 2 diabetes is a heterogeneous disease, and HIIT's effects may vary depending on factors like disease duration, insulin use, and sedentary lifestyle. What is the theoretical basis for suggesting that increased endogenous antioxidant capacity improves redox balance in these patients? What do multivariate analyses indicate? Could HIIT improve parameters like HbA1c independently of oxidative stress markers?

Response:

I appreciate this valuable comment and thank you for these important suggestions. Accordingly, I have revised the manuscript at several places in this Section to provide a more nuanced perspective. Please see, page 7-8, lines 339-359:

„From a mechanistic standpoint, recent research highlights the crucial role of the Nrf2 pathway in mediating the beneficial redox adaptations to exercise by activating endogenous antioxidant defenses and promoting cellular protection against oxidative stress. Regular (particularly higher-intensity) exercise stimulates Nrf2 activity, leading to an increase in the transcription of genes encoding antioxidant and detoxification enzymes that combat ROS produced during physical activity [49]. Additionally, Nrf2 activation has been shown to improve mitochondrial efficiency and glucose uptake [49], offering a plausible pathway by which enhanced endogenous antioxidant capacity re-stores redox balance and supports metabolic control in T2DM.“

„In line with this, Kazemi et al. [50] demonstrated that a 12-week cycle ergometer-based HIIT intervention led to significant increases in H2O2 and Nrf2 levels in male patients with T2DM compared to controls. Concurrently, antioxidant enzyme levels (particularly CAT) were significantly elevated in the HIIT group. Moreover, HIIT im-proved various metabolic parameters (e.g., 12 h fasting plasma glucose, HbA1c, blood lipids). These findings suggest that although HIIT acutely raises oxidative stress, it also activates protective antioxidant pathways (likely via Nrf2 signaling) in T2DM patients.“

„However, improvements in clinical outcomes (e.g., HbA1c) following HIIT have also been found to occur independently of changes in oxidative stress biomarkers. A recent metaanalysis of 22 randomized controlled HIIT trials reported significant im-provements in glucose and lipid metabolism markers in T2DM cohorts [51], and it has been demonstrated that several exercise-related mechanisms can contribute independently to improved glycemic control, including enhanced glucose transporter type 4 (GLUT4) translocation [52], increased muscle capillarization [53], and improved hepatic insulin signalling [52]. Consequently, endogenous antioxidant adaptation should be viewed as a complementary rather than singularly causal factor in the metabolic bene-fits conferred by HIIT. Additionally, it is important to note that T2DM is a heterogeneous disorder in which HIIT-related adaptations may vary with disease duration, pharmacotherapy (particularly insulin use), and habitual physical activity. It has been reported, for example, that shorter disease duration (< 5 years) and an age of <60 years predict larger glucose and lipid metabolism in T2DM patients following HIIT interven-tions, whereas longstanding disease cases and older patients (>60 years) exhibit attenuated responses [51].“

#5

The table should separate the main effects column into 2 columns: one indicating markers of oxidative stress, and the other one describing the parameters of disease improvement (glycemic control, blood pressure, blood lipids).

Response:

Thank you for this suggestion. I have revised the table accordingly.

#6

The conclusion lacks support due to the mixture of study designs and lack of systematization.

Response:

Once again, I thank you for this valuable recommendation. Accordingly, I have revised the Conclusions and Perspectives section at several places. Please see, page 13, lines 515-532:

„However, these findings should be considered in the context of a still heterogeneous and evolving body of evidence. Many conclusions are derived from small-to-moderate-sized randomized controlled or quasi-experimental studies that vary widely in design, intervention protocols, study duration, and biomarker selection. Moreover, the mechanistic contribution of oxidative stress modulation to clinical im-provements remains context-dependent. In some studies, changes in redox biomarkers occurred alongside metabolic and vascular benefits, while in others, clinical gains were also evident in the absence of measurable oxidative adaptations. This suggests that redox modulation may serve a complementary, rather than singularly causal, role in the overall efficacy of HIIT.“

„Importantly, age, hormonal status, and individual redox responsiveness appear to influence outcomes. For instance, in older adults and postmenopausal women, vascular improvements have been observed without significant alterations in skeletal muscle oxidative biomarkers, potentially reflecting diminished adaptive plasticity. Nonetheless, these populations still exhibit clinically meaningful benefits in cardiometabolic indices such as decreased blood pressure, indicating that HIIT can confer value despite an unchanged redox status. While some studies suggest that higher-volume HIIT protocols may yield greater redox benefits, lower-volume regimens have also been shown to induce protective changes.

page 14, lines 554-570:

„Looking ahead, several avenues warrant further exploration to better harness HIIT as a redox-modifying therapy. First, given the mixture of study designs and study populations, future studies should aim to investigate interindividual variability by stratifying for age, duration and severity of disease, sex hormone status, habitual physical activity patterns, dietary intake, and concurrent pharmacotherapy. Second, there is a need to explore the interaction between training intensity, volume, and program duration in determining redox outcomes. Third, longer-lasting studies are needed to assess the long-term sustainability of redox improvements and their predictive value for clinical endpoints such as cardiovascular events and disease progression. For in-stance, while there is emerging evidence that certain cardiometabolic adaptations following HIIT may persist beyond the active training period [68–70], further research is warranted to determine whether redox adaptations show similar persistence during periods of detraining. Finally, the potential synergistic effects of combining HIIT with adjunctive strategies, such as targeted nutrition or pharmacological agents, merit deeper exploration. As the body of research on HIIT and redox biology in cardiometabolic diseases continues to grow, further well-designed, hypothesis-driven studies and future systematic reviews will be essential to quantitatively assess intervention effects and guide clinical translation.“

Reviewer 3 Report

I appreciate the opportunity to review this manuscript. It offers an interesting narrative review on the effects of high-intensity interval training on oxidative stress and antioxidant capacity across various cardiometabolic disease cohorts.  

The review is well-structured and well-written, offering an extensive literature overview. Results are effectively presented in a clarifying table. This document could serve as valuable reference material for students, educators, researchers, and clinicians.

My proposed corrections are minor, as follows.

Line 91. Could you please add a little more information about ROS and mitochondrial impairment?

Line 225. “Repeated stimulation of this pathway by HIIT results in sustained upregulation of cellular defenses and enhanced metabolic resilience, even in the absence of further training.” Could you add, if available, a little more information about the duration of follow-ups studying this sustained upregulation of cellular defenses and enhanced metabolic resilience?

Line 312. “fasting glucose)”. Please, add the period.

Table 1. According to journal recommendations it is necessary to add in the legend of the table the explanation of the acronyms used in the table.

Line 444. Could you add a reflection comparing the benefits of HIIT in relation with MICT? Why HIIT may be superior and in which parameters specifically? Because MICT may be superior, for example, in favoring weight loss, and thus allowing maybe for more sustainable changes in time. The potential of protocols combining both HIIT and MICT, especially in the initial training, may be highlighted.

Author Response

Major comments

I appreciate the opportunity to review this manuscript. It offers an interesting narrative review on the effects of high-intensity interval training on oxidative stress and antioxidant capacity across various cardiometabolic disease cohorts.

The review is well-structured and well-written, offering an extensive literature overview. Results are effectively presented in a clarifying table. This document could serve as valuable reference material for students, educators, researchers, and clinicians.

Response:

I sincerely thank you for the positive and encouraging evaluation of my manuscript. I am grateful for your thoughtful feedback and greatly appreciate your recognition of the manuscript’s potential impact to the scientific and clinical community. I am pleased that the review was found to be of value for a broad audience. I have carefully addressed your specific suggestions below to further strengthen the manuscript.

Comment #1

My proposed corrections are minor, as follows.

Line 91. Could you please add a little more information about ROS and mitochondrial impairment?

Response:

I thank you for this insightful suggestion and I have revised the text accordingly to provide a clearer description of how ROS contribute to mitochondrial dysfunction in cardiometabolic diseases. Please see, page 3, lines 102-107:

„Excessive ROS generation can also directly impair mitochondrial function by damaging mitochondrial DNA, lipids, and respiratory chain proteins, leading to decreased ATP production, increased electron leakage, and further ROS generation, a vicious cycle contributing to metabolic inflexibility and cellular energy deficits [16]. Such mitochondrial dysfunction is particularly detrimental in insulin-sensitive tissues like skeletal muscle and the heart, where it exacerbates insulin resistance and impairs con-tractile function.“

Comment #2

Line 225. “Repeated stimulation of this pathway by HIIT results in sustained upregulation of cellular defenses and enhanced metabolic resilience, even in the absence of further training.” Could you add, if available, a little more information about the duration of follow-ups studying this sustained upregulation of cellular defenses and enhanced metabolic resilience?

Response:

Thank you for your insightful suggestion. Given an apparent lack of longer-term follow-up studies, I have revised the original statement to better reflect the current evidence base by tempering the claim regarding the sustained upregulation of „cellular defenses“ after HIIT. Specifically, I have deleted this paragraph and, instead, I have provided some discussion to Perspectives section to clarify that there is emerging evidence supporting the persistence of certain cardiometabolic benefits following HIIT detraining but that there is currently a lack of long-term follow-up studies directly examining whether redox-related adaptations are sustained after cessation of HIIT. Please see, page 14, lines 560-565:

„[…] longer-lasting studies are needed to assess the long-term sustainability of redox improvements and their predictive value for clinical endpoints such as cardiovascular events and disease progression. For instance, while there is emerging evidence that certain cardiometabolic adaptations following HIIT may persist beyond the active training period [68–70], further research is warranted to determine whether redox ad-aptations show similar persistence during periods of detraining.“

Comment #3

Line 312. “fasting glucose)”. Please, add the period.

Response:

Thank you for this suggestion. I have added the time period (8 h).

Comment #4

Table 1. According to journal recommendations it is necessary to add in the legend of the table the explanation of the acronyms used in the table.

Response:

Thank you for drawing my attention to this. Accordingly, I have included all explanations of the acronyms used in the table legend.

Comment #5

Line 444. Could you add a reflection comparing the benefits of HIIT in relation with MICT? Why HIIT may be superior and in which parameters specifically? Because MICT may be superior, for example, in favoring weight loss, and thus allowing maybe for more sustainable changes in time. The potential of protocols combining both HIIT and MICT, especially in the initial training, may be highlighted.

Response:

I thank you once again for this thoughtful suggestion. I absolutely agree that contextualizing HIIT relative to MICT more explicitly enhances the review’s practical impact. Accordingly, I have now added a discussion segment to address your comment properly. Please see, page 13, lines 535-548:

When comparing HIIT to more traditional MICT prescriptions, research has con-sistently show that both modalities can produce significant reductions in body mass and body composition, while HIIT appears to offer greater improvements in VO₂max and some cardiometabolic risk indices [9,27,28,64]. Importantly, improved VO₂max (the key marker of cardiorespiratory fitness) is strongly associated with reduced mortality risk [65], and mitochondrial enzyme activities (e.g., citrate synthase and complex I) have been reported to increase more after HIIT compared to MICT, demonstrating more pronounced mitochondrial adaptation [66]. However, given a dose-response association between the amount of physical activity and weight loss [67], higher-volume MICT may be more effective for sustainable body weight reduction. Thus, exercise programs that integrate both HIIT and MICT may offer synergistic benefits (e.g., HIIT to rapidly enhance cardiorespiratory fitness and mitochondrial capacity in a more time-efficient and feasible way, and MICT to support sustained weight management). This blended strategy deserves further exploration in future clinical trials.“

Round 2

Reviewer 2 Report

The changes made by the author were satisfactory. There are some typo errors that will need to be review es in the final edit.

No more comments.